# First fully-automated AI/ML virtual screening cascade implemented at a drug discovery centre in Africa

Gemma Turon [1,4], Jason Hlozek [2,4], John G. Woodland [2,3], Ankur Kumar [1], Kelly Chibale [2,3] ✉ & Miquel Duran-Frigola [1] ✉

Streamlined data-driven drug discovery remains challenging, especially in resource-limited settings. We present ZairaChem, an artificial intelligence (AI)- and machine learning (ML)-based tool for quantitative structure-activity/ property relationship (QSAR/QSPR) modelling. ZairaChem is fully automated, requires low computational resources and works across a broad spectrum of datasets. We describe an end-to-end implementation at the H3D Centre, the leading integrated drug discovery unit in Africa, at which no prior AI/ML capabilities were available. By leveraging in-house data collected over a decade, we have developed a virtual screening cascade for malaria and tuberculosis drug discovery comprising 15 models for key decision-making assays ranging from whole-cell phenotypic screening and cytotoxicity to aqueous solubility, permeability, microsomal metabolic stability, cytochrome inhibition, and cardiotoxicity. We show how computational profiling of compounds, prior to synthesis and testing, can inform progression of frontrunner compounds at H3D. This project is a first-of-its-kind deployment at scale of AI/ML tools in a research centre operating in a low-resource setting.

The cost of bringing new medicines from the bench to the bedside has risen steadily since the 1970s[1]. Recent estimates suggest a median cost of $1.3 billion per drug[2] with research and development taking an average of 10 years[3]. To avoid costly failures, the drug discovery industry has turned to artificial intelligence (AI) and machine learning (ML) to accelerate research timelines and reduce attrition rates, with investments in AI/ML soaring in the last five years[4]. The application of AI/ML aims to transform drug discovery from a slow, sequential, high-risk process to a fast, finely-tuned and integrated pipeline that expedites the delivery of novel clinical candidates with reduced risk of failure[5]. Amongst other applications, AI/ML is now embedded within quantitative structure-activity/property relationship (QSAR/QSPR) modelling, boosting performance and providing additional insights for drug design[6,7].

The promise of AI/ML for biomedicine extends to the field of infectious diseases, which are currently underrepresented in drug discovery portfolios[8]. Infectious diseases predominantly afflict lower-to-middle-income countries (LMICs), most of which are situated in the Global South. For example, Africa carries over 95% of the 240 million annual global cases of malaria[9] and 25% of global deaths from tuberculosis[10]. Historically, efforts to tackle these challenges have principally occurred in the Global North; consequently, African drug discovery efforts have largely been dependent on international funding agencies with programmes driven from abroad. AI/ML methods offer an opportunity to revitalise and expedite drug discovery projects conducted in low-resource settings; however, a lack of data science expertise and limited access to computational resources hinder the uptake of AI/ML at research institutions and universities in LMICs[11]. It is

[1]Ersilia Open Source Initiative, Cambridge, UK. [2]Department of Chemistry and Holistic Drug Discovery and Development (H3D) Centre, University of Cape Town, Cape Town, South Africa. [3]South African Medical Research Council Drug Discovery and Development Research Unit, Institute of Infectious Disease and Molecular Medicine, University of Cape Town, Cape Town, South Africa. [4]These authors contributed equally: Gemma Turon, Jason Hlozek. ✉e-mail: kelly.chibale@uct.ac.za; miquel@ersilia.io

anticipated that lowering these barriers may lead to important scientific contributions from those countries that disproportionately suffer from the bulk of infectious diseases, a milestone towards their eradication.

Since its launch in 2010, the Holistic Drug Discovery and Development (H3D) Centre at the University of Cape Town in South Africa has made significant advances in innovative drug discovery projects and infrastructure development, intimately aligned with capacity strengthening across the African continent[12]. This includes the discovery of the first-ever small-molecule clinical candidate, for any disease, researched on African soil by an international team led by an African drug discovery centre[13] which subsequently reached Phase II human trials in African malaria patients. To advance its mission of discovering and developing novel, life-saving medicines for infectious diseases that predominantly affect African populations, H3D works closely with the Ersilia Open Source Initiative (EOSI), a non-profit organisation aimed at disseminating AI/ML methodologies applied to urgent biomedical needs in LMICs.

Here we describe ZairaChem, an automated pipeline for AI/ML-based QSAR/QSPR modelling, designed for fast and easy implementation in low-resource settings. We demonstrate the application of ZairaChem to key assays in the antimalarial and antitubercular drug discovery programs conducted at H3D. The AI/ML models are arranged in the form of a virtual screening cascade that mirrors the progression of compounds in a 'real-world' experimental setting. Virtual screening is a critical tool in drug discovery, allowing for the identification of new hits and the prioritisation of compounds for testing, resulting in a significant reduction in experimental attrition rates. The developed AI/ML assets include models corresponding to whole-cell phenotypic screening assays against *Plasmodium falciparum* (*Pf*) and *Mycobacterium tuberculosis* (*Mtb*), as well as common absorption, distribution, metabolism, excretion and toxicity (ADMET) assays such as aqueous solubility, microsomal metabolic stability and cardiotoxicity.

This is the first comprehensive AI/ML-based QSAR/QSPR virtual screening cascade that, to our knowledge, has been brought to production in a drug discovery setting on the African continent. ZairaChem can run on conventional computers and is fully automated, requiring limited data science expertise and allowing for periodic model updates with new data. We believe that this virtual screening cascade, based on in-house data and free open-source software, has the potential to set the basis for sustainable, affordable and scalable drug discovery initiatives in the Global South.

## Results

### Available data and screening cascades at H3D

We have modelled two virtual drug discovery cascades focused on identifying potential novel antimalarial and antituberculosis compounds and their ADMET properties. We have selected assays representative of each step of the experimental drug screening cascade and for which H3D had sufficient available in-house data (i.e., at least 100 molecules), viz. whole-cell screening data against *P. falciparum* (NF54 and K1 strains) and *M. tuberculosis* (H37Rv strain), cytotoxicity against two mammalian cell lines (CHO, HepG2), aqueous solubility, microsomal metabolic stability in human, mouse, and rat liver microsomes, and permeability (Caco-2). Key assays for compound progression in the cascade but for which only a small subset of data was available (e.g., human cytochrome P450 (CYP) inhibition and hERG blockade) have been developed using publicly-available data (see Materials and Methods). Advanced drug metabolism and pharmacokinetics (DMPK) assays have not been included in the virtual screening cascade at this stage of implementation. Supplementary Table 1 lists a brief description of each assay, the number of compounds available, and cut-offs used to obtain binary outcomes (1/0; active/inactive, soluble/not soluble, etc.). Activity cut-offs were determined through consultation

with experts at H3D to ensure the relevance of model predictions for projects across different disease areas.

### Automated AI/ML modelling with ZairaChem

To streamline model development and facilitate adoption and maintenance of the AI/ML assets at H3D, we developed ZairaChem, an automated AI/ML tool to train classification models able to predict the probability of "1" (typically an "active" assay outcome) of new compounds, given only their chemical structures represented as SMILES strings (Fig. 1a, Materials and Methods). In brief, molecules are represented numerically using a combination of distinct descriptors, including physicochemical parameters (Mordred[14]), 2D structural fingerprints (ECFP[15]), inferred bioactivity profiles (Chemical Checker[16]), graph-based embeddings (GROVER[17]), and chemical language models (ChemGPT). The rationale is that combining multiple descriptors will enhance applicability over a broad range of tasks, ranging from aqueous solubility predictions to phenotypic outcomes. Subsequently, a battery of AI/ML algorithms is applied using modern automated machine learning (AutoML) techniques aimed at yielding accurate models without the need for human intervention (i.e., algorithm choice, hyperparameter tuning, etc.). The AutoML frameworks FLAML[18], AutoGluon[19], Keras Tuner[20] and TabPFN[21] were incorporated, covering mostly tree-based methods (random forest, XGBoost, etc.) and neural network architectures.

In order to demonstrate the applicability of the ZairaChem pipeline to a broad chemical space as well as to a variety of prediction tasks, we benchmarked the tool in the Therapeutics Data Commons ADMET binary classification tasks[22]. Out-of-the-box, ZairaChem models demonstrated state-of-the-art (SOTA) performance across all classification tasks, scoring between 1st and 4th in all benchmark datasets (Supplementary Table 2). Furthermore, we compared the prediction performance of our in-house activity models against models from the literature to assess the potential impact of our tool at H3D over existing tools (Supplementary Fig. 1). These external tools include predictors for bioactivity available from the Ersilia Model Hub; namely, a version of the malaria inhibition prediction (MAIP)[23] model for *P. falciparum* and ChemTB[24] for *M. tuberculosis*, as well as ADME@NCATS[25,26] suite for ADME predictions for solubility, microsomal stability in rat and human, and CYP inhibition (CYP2C9, CYP2D6, CYP3A4). While all models provide hit enrichment within the H3D chemical space, the ZairaChem pipeline enables the training of in-house models with internal data for greater predictive power.

Figure 1 exemplifies ZairaChem applied to the H37Rv strain of *Mtb*. As of November 2021, 3,244 molecules had been screened in this assay at H3D, spanning a diverse chemical space (Fig. 1c). In total, 81 chemical series are represented, with 20 series covering 80% of the molecules. Setting an $MIC_{90}$ cut-off of 5 μM yields 483 actives available for training the AI/ML model (Fig. 1b). 20% of the data was held as a test set. An ensemble of models was fitted based on the multiple small-molecule descriptors, and performance was evaluated for each model individually (Fig. 1f). The outcome of models inside the ensemble was aggregated in a consensus score estimating the probability of observing an "active" (1) assay outcome. Indeed, in the hold-out test set, known active molecules scored higher than known inactive compounds (Fig. 1d). Measured in the receiver operating characteristic (ROC) space, the consensus score achieves an area under the ROC curve (AUROC) of 0.92, higher than any of the individual classifiers alone. In this case, a default score threshold of 0.5 can be established to classify predictions as "actives" (1) and "inactives" (0). Based on this threshold, the model predicted 90 active molecules from the test set, of which 71 were true positives, corresponding to a precision of 78.9% and a recall of 66.4% (Fig. 1h).

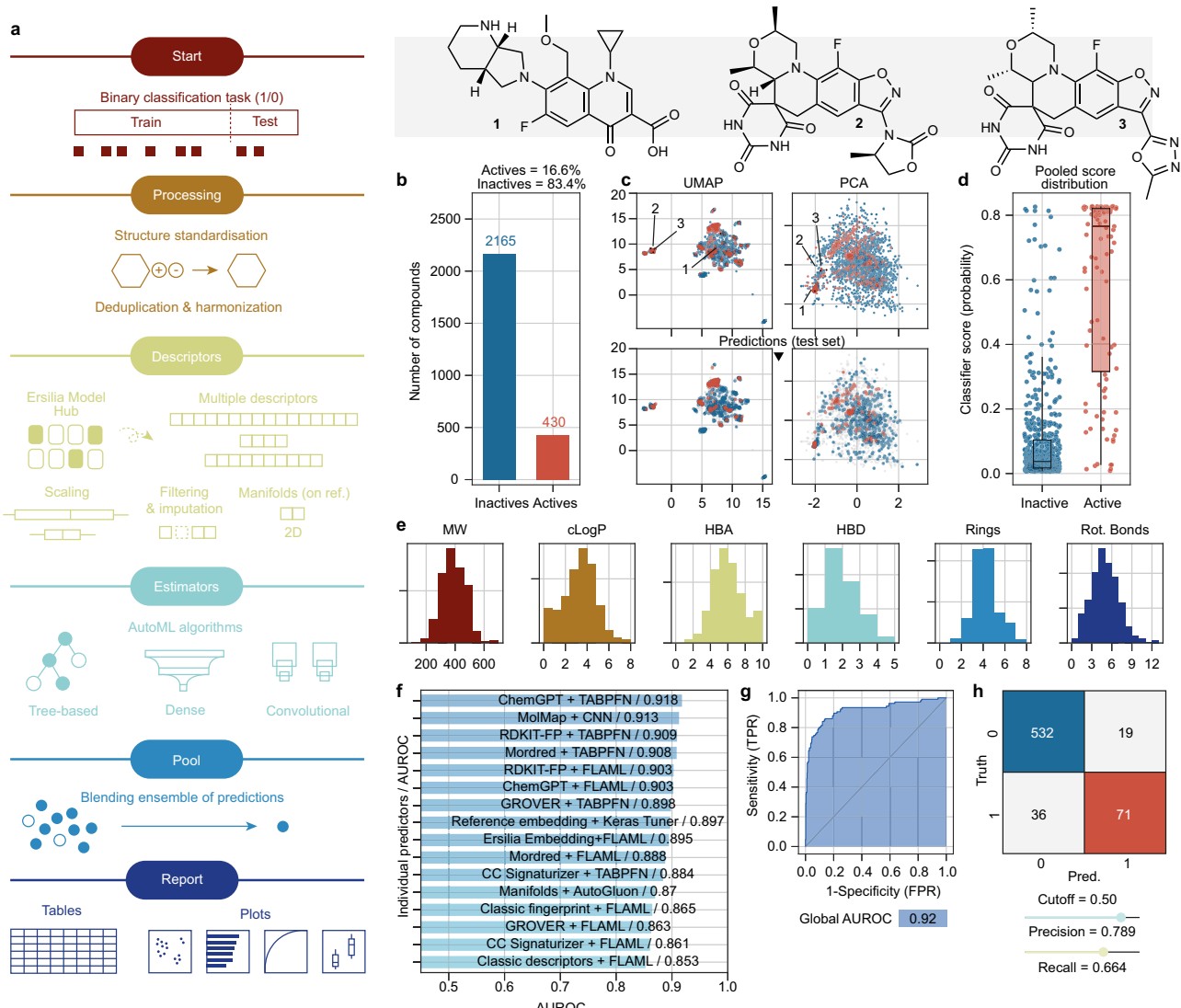

**Fig. 1 | The ZairaChem pipeline. a** Scheme of the AutoML methodology, consisting of data processing, descriptor calculation, training of models, assembling (pooling) of results, and reporting. **b** Number of active and inactive compounds in the *Mtb* MIC90 assay (training set). **c** Uniform manifold approximation and projection (UMAP) and principal component analysis (PCA) projections of the chemical space in the *Mtb* MIC90 assay. Structurally different (1 vs 2/3) and similar (2 vs 3) compounds are depicted. Red indicates active compounds; blue indicates inactive compounds. **d** Model scores (probability of "1") assigned to the true active (red, $n = 107$) and inactive (blue, $n = 542$) compounds in the test set (20% of the total available data). Boxes indicate the median (central line), Q1 (upper bound) and Q3

(lower bound), and whiskers extend to the data points within up to 1.5 times in the interquartile range. **e** Distribution of common chemical properties of the compounds, namely molecular weight (MW), calculated logP (cLogP), number of hydrogen bond acceptors (HBA), number of hydrogen bond donors (HBD), number of rings (Rings) and number of rotatable bonds (Rot. Bonds). **f** AUROC scores of the individual ZairaChem predictors. **g** ROC curve of the final ensemble model. **h** Confusion matrix showing true positives (red), true negatives (blue), false positives, and false negatives in the test set. Source data are provided as a Source Data file.

## Systematic AI/ML modelling of H3D screening cascades

We applied the ZairaChem pipeline to the experimental assays available at H3D (Fig. 2a and Supplementary Table 1). The resulting models showed good performance (AUROC > 0.7, Fig. 2b) and well-scaled prediction scores within the [0-1] range for all in-house datasets: *Pf* NF54, *Pf* K1, *Mtb* H37Rv, CHO, HepG2, aqueous solubility (Aq Sol), Caco-2 permeability (Caco-2) and intrinsic clearance ($CL_{int}$) for human (H), mouse (M), and rat (R) microsomes (Fig. 2c, Supplementary Fig. 2 and Table 3). We observed that ZairaChem classifiers successfully up-rank active compounds (Fig. 2d), with significant enrichment of hits within the top 50 candidates (Fig. 2e). We found similar performance for the remaining assays developed with in-house data beyond those depicted in Fig. 2 (Supplementary Fig. 2 and 3).

Data points were scarce for key assays related to more advanced stages of the screening cascade. Experiments related to drug metabolism and off-target binding, such as interactions with cytochrome P450 enzymes (CYPs) or inhibition of the hERG ion channel, are costly and often not performed on-site for many drug discovery organisations. In the case of CYPs, we gathered bioactivity data for over 15,000 molecules available from the PubChem BioAssay[27] and ChEMBL databases[28]. We built AI/ML models for the CYP3A4, CYP2C9, CYP2C19 and CYP2D6 isoforms. Remarkably, except for CYP2C19, CYP models built with public data were able to achieve good performance (AUROC > 0.65) in the H3D chemical space (Fig. 2b, d, Supplementary Fig. 3, Table 3), assigning high scores to active ("1", CYP inhibitors) and low scores to inactive compounds ("0", no CYP inhibition observed)

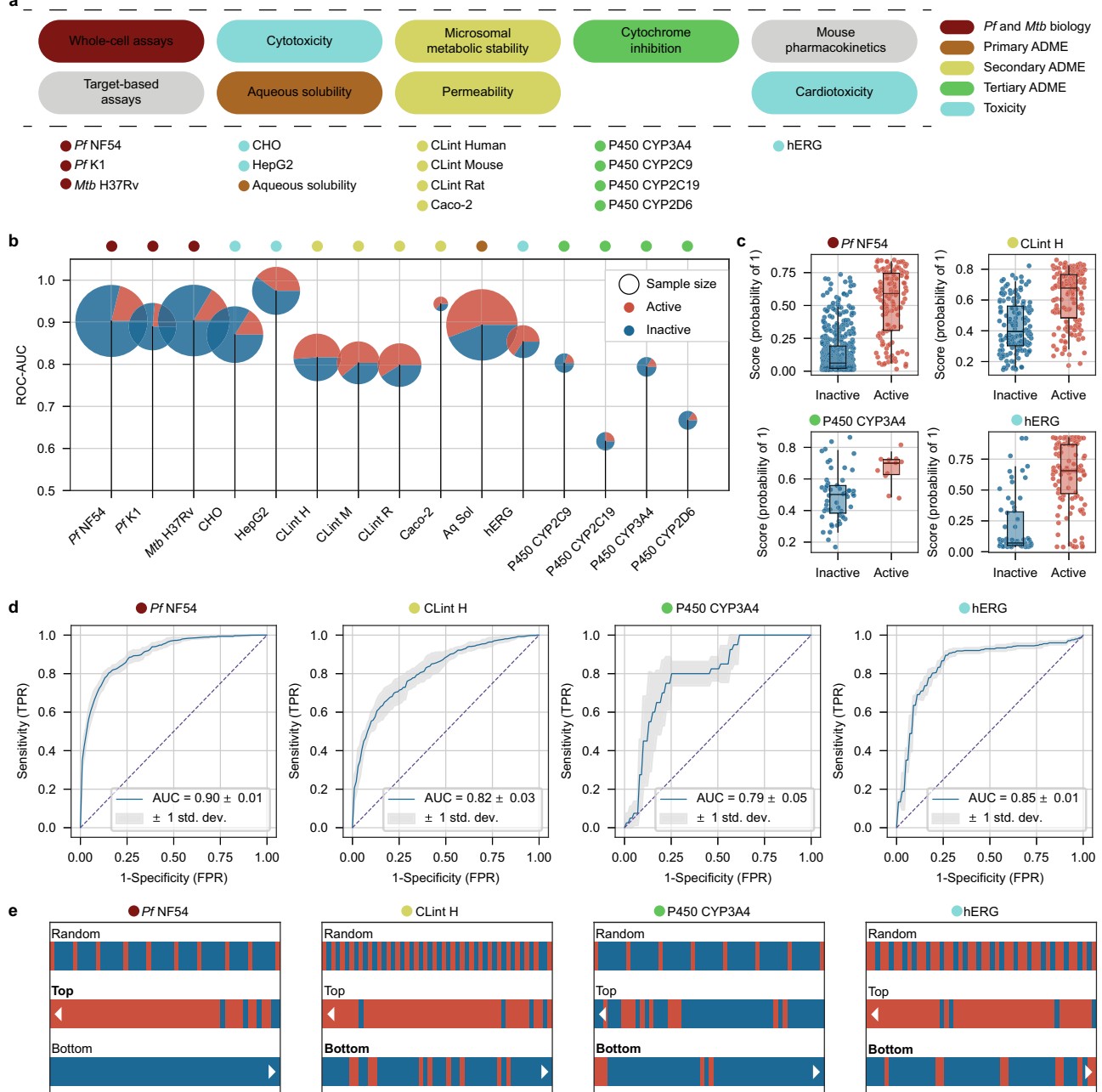

**Fig. 2 | ZairaChem implementation of a virtual screening cascade. a** Summary of assays most frequently used in drug discovery programmes at the H3D Centre, progressing from left to right. **b** AUROC for the 10 AI/ML models developed with internal data (test sets: 20% of H3D data), the four CYP models developed with ZairaChem using external data, and the CardioToxNet model from the literature (test sets: 100% of H3D data). Dataset sample counts are represented by circle size with the corresponding proportion of active (red) and inactive (blue) compounds. **c** Classification scores of individual compounds for representative stages of the screening cascade. N. Active/inactive: NF54 139/519, ClintH 146/ 140, CYP3A4 8/41, hERG 90/47. Boxes indicate the median (central line), Q1 (upper

bound) and Q3 (lower bound) and whiskers extend to the data points up to 1.5 times in the interquartile range. **d** Correspondingly, ROC curves resulting from a five-fold cross-validation, with blue lines depicting the mean AUROC. **e** Comparison of hit rates for randomly selected molecules (first row) vs molecules ranked according to the model score (probability of "1") for selected assays. The top 50 (second row) and bottom 50 (third row) molecules are depicted, showing a hit enrichment of true active compounds (red) in the highest-ranked positions and an enrichment of true inactive compounds (blue) in the lowest-ranked positions. Source data are provided as a Source Data file.

(Fig. 2c and Supplementary Fig. 2). These models successfully enable the selection of compounds that are not likely to interact with the selected CYPs (10 bottom-ranking compounds; Fig. 2e, Supplementary Fig. 4 and Table 4).

In addition to publicly-available datasets, pre-trained AI/ML models are becoming more frequent in the scientific literature. The core mission of EOSI is to collect such public models in a unified, easy-

to-use repository named the Ersilia Model Hub[29] (https://ersilia.io/ model-hub). As of May 2023, the Ersilia Model Hub contains over one hundred models for drug discovery, with a focus on infectious disease research. To demonstrate the potential of this resource, we chose to fetch a hERG blockade prediction model, corresponding to CardioToxNet[30] and also available through the Ersilia Model Hub. This model was used "as is", without further fine-tuning with H3D data, and

showed excellent accuracy (AUROC = 0.852) on H3D compounds (Fig. 2b, d, Supplementary Table 3), with good discriminative scores between active (1, cardiotoxic) and inactive compounds (0, non-cardiotoxic) (Fig. 2c), as well as an enrichment of inactivity in the bottom-ranked 50 compounds (Fig. 2e).

Furthermore, to measure the advantage of using AI/ML models to prioritise compounds for experimental screening, we have calculated the hit enrichment potential of each model. Overall, assays where an "active" outcome is desired (*Pf* NF54, *Pf* K1, *Mtb* H37Rv, Aq Sol, and Caco-2) show between 25% and 70% hit rate improvement in the top 50 molecules. For example, out of 50 randomly-selected molecules in the *Pf* NF54 dataset, by chance, 10 would be active (20%); however, if we rank the data according to the AI/ML model scores, 45 molecules (90%) are found to be active ($IC_{50} < 0.1\,\mu M$). Assays where the desired outcome is "inactivity" show a hit rate improvement for inactive molecules of 18% to 46% in the bottom 50 molecules (for cytotoxicity, clearance and hERG models) or bottom 10 molecules (for CYP P450 models) as ranked by the AI/ML model score (Supplementary Table 3, Supplementary Figs. 4 and 5).

Finally, we demonstrate the ability of the ZairaChem pipeline to facilitate transference from literature data towards a narrower chemical space. We show how models trained on external datasets (P450 CYPs models) can improve their performance when in-house data points are included in the training set. Notably, the AUROC of the worst-performing models on H3D data (CYP2C19 and CYP2D6) both

improve upon the addition of internal data points (Supplementary Fig. 6).

## AI/ML performance by chemical series

In the early stages of a drug discovery project, derivatives of a compound which retain the structural core, or pharmacophore, are designed and synthesised to optimise the biochemical and bioactivity properties of that chemical series. Focusing on *Mtb* and *Pf* bioactivity prediction, we investigated how many molecules of a given chemical series were needed to illuminate that space. We trained sets of models on bioactivity data and gradually increased the number of training points from specific chemical series to observe the effect of increasing "local" training data on model performance. As expected, the predictive potential for a chemical series improved with an increase in local data density, where approximately 30 molecules from a given series provide a good starting point to produce predictive models (Fig. 3c, g). Secondly, we measured the impact of the availability of global data on model quality at the chemical series level (Fig. 3d, h) by comparing models trained on 100 series-specific compounds alone to training sets that also included the broader *Pf* and *Mtb* H3D libraries. In general, the addition of more data, even if corresponding to a more general chemical space, improves model performance for analogues within a chemical series. Series 2 of *Mtb*, the exception to this trend, had very few active compounds available for training and is also structurally distinct from the remainder of the H3D library for *M.*

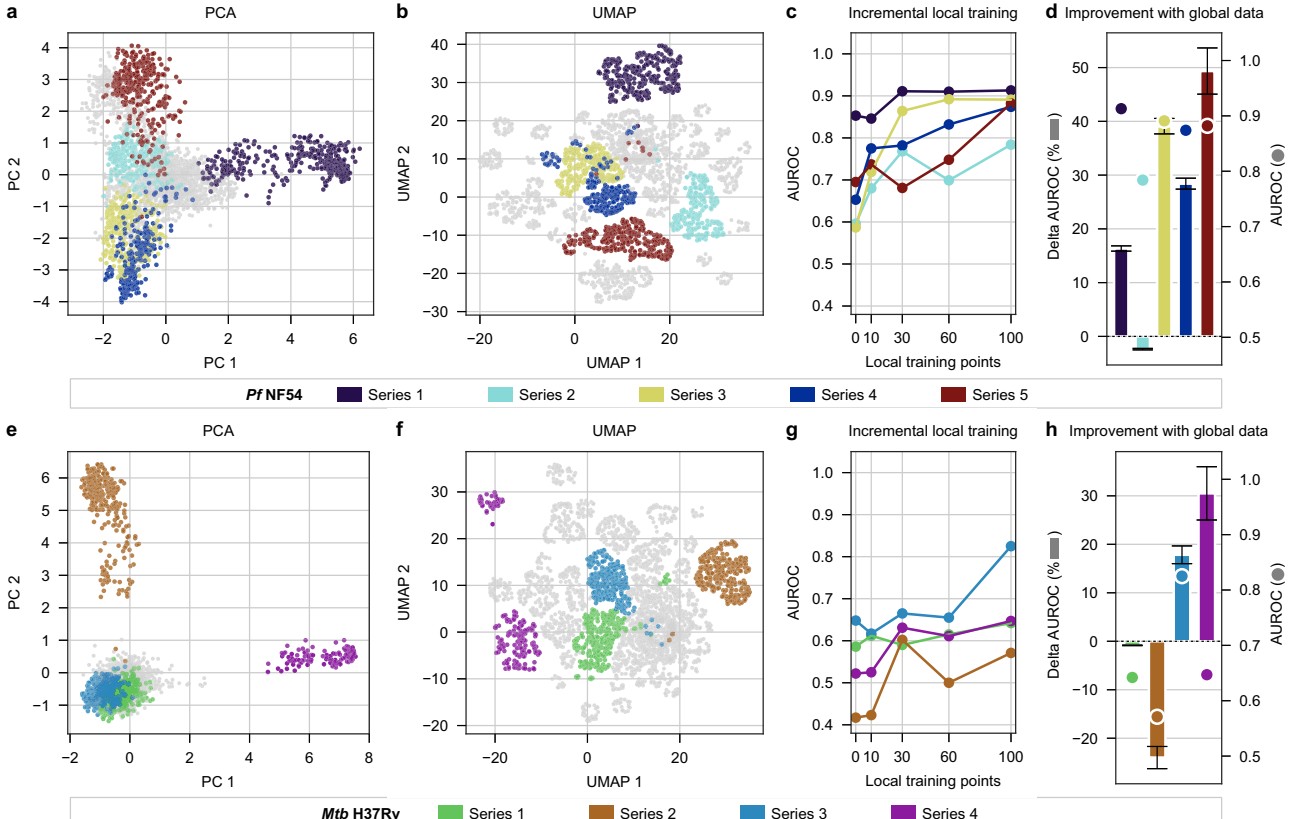

**Fig. 3 | Model performance within chemical series corresponding to novel regions of chemical space.** PCA and UMAP projections of the chemical space of the H3D Centre's library for specific chemical series in the malaria (top row) and tuberculosis (bottom row) disease areas. **a**, **e** PCA preserves the global distribution of chemical space while **b**, **f** UMAP emphasises the clustering of structurally similar data points. **c**, **g** Median AUROC scores from a five-fold cross-validation are measured for training sets with an incremental number of local training points for each series, respectively. **d**, **h** The percentage of change towards a perfect model

(AUROC = 1) between a model trained on a dataset that includes compounds from a more general chemical space versus a model trained on series-specific data alone (see calculation in Methods). The median AUROC score from a five-fold cross-validation, for models trained with both 100 series-specific compounds and global data, is plotted with a circle corresponding to the values of the right-hand-side y-axis. Error bars indicate ± standard deviation ($n = 5$). Source data are provided as a Source Data file.

*tuberculosis*, leading to poor model performance that the addition of global data could not rescue.

## Application to an unseen collection of compounds

To demonstrate the effectiveness of the virtual screening cascade for de novo screening of molecular libraries, we reproduced the discovery of a potent antiplasmodium compound at H3D from a series of 2,4-disubstituted imidazopyridines[31]. In this study, the authors identified two initial hits with moderate antiplasmodium activity against asexual blood stage parasites ($IC_{50}$ = 0.24 μM and 0.49 μM, respectively) and derived a series of 65 compounds for experimental testing. All molecules were tested against asexual blood stage drug-sensitive (NF54) and drug-resistant (K1) *Pf* strains. Their cytotoxicity in CHO cells as well as aqueous solubility (pH 6.5) were experimentally determined. In addition, some molecules progressed further in the cascade and were tested for cardiotoxicity risk (hERG blockade) and microsomal metabolic stability (Mouse $CL_{int}$). To validate the models in the context of this series, we removed these compounds from the training sets of the relevant AI/ML models in the virtual screening cascade (*Pf* NF54, *Pf* K1, CHO, Aq Sol at pH 6.5, Caco-2, $CL_{int}$ Human, Rat, and Mouse). ZairaChem models show good performance (AUROC > 0.75) on the experimentally validated library (Fig. 4a). Next, we leveraged the model scores to create visual fingerprints for ease of identification of potential hits: those with high scores for the desired activities (dark red) and low scores for the undesired activities (light blue) (Fig. 4b). We selected a few molecules that showed the desired pattern (Fig. 4c) and compared their predicted activity from the ZairaChem models with the experimental results (Fig. 4d). Compound 1, the initial hit of the series, is also shown for reference. We demonstrate that AI/ML models allow for the selection of candidates with high chances of progression in the cascade (compounds 37 and 55) and, conversely, molecules with undesired side effects could be discarded prior to experimental testing, despite their high bioactivity against *Pf* (compounds 22 and 58). Indeed, compound 37, one of the molecules selected by our model predictions for its high activity against *Pf* and low toxicity profile, was the lead compound from this study, showing in vivo efficacy in the humanised SCID NSG mouse malaria infection model.

## Application to active research projects

Finally, to demonstrate the impact of these AI/ML models on the drug discovery pipeline at H3D and to highlight the capacity of these tools to facilitate the identification of lead-like compounds, we applied the virtual screening cascade to two active medicinal chemistry programmes in a 'real-world' prospective study (Fig. 4e) following model deployment. These programmes, in the lead optimisation and hit-to-lead stages, respectively, are represented by the naphthyridine and pyrazole chemotypes (Fig. 4f) which target the *Pf* phosphatidylinositol 4-kinase and mycobacterial membrane protein large 3 transporter, respectively. To build confidence in our model predictions before adoption as a tool to direct chemical synthesis efforts, all new derivatives from both chemical series were assessed using our virtual screening cascade. We compared the model scores of the previously-unseen compounds to the corresponding experimental values for the assays once experimental data were obtained. Predictions for the novel naphthyridine and pyrazole compounds from the activity models (*Pf* NF54 and *Mtb*, respectively) and solubility classifiers (at pH 6.5 and 7.4, respectively) showed broad agreement (green) with the respective measured assay values, with only limited instances where the model produced a confident yet incorrect prediction (dark purple). The corresponding swarm plots (Fig. 4g) show good separation between confirmed actives and inactives, illustrating how our AI/ML models can efficiently prioritise novel compounds and facilitate faster elucidation of structure-activity relationships in a resource-constrained environment.

Indeed, in terms of precision (P) and recall (R) at a permissive prediction cut-off (active if > 0.3), we found, for the naphthyridine series: *Pf* NF54 P = 0.333, R = 0.529; solubility P = 0.648, R = 0.946; and, for the pyrazole series: *Mtb* P = 0.577, R = 0.872; solubility P = 0.727, R = 1.0; further model performance metrics at different prediction cut-offs are presented in Supplementary Table 5. In addition, models showed good prospective performance across broader H3D chemical space when validated on all novel compounds synthesised in the year following model development (Supplementary Fig. 7). Therefore, this 'real-world' model validation demonstrates the ability of our AI/ML models to identify analogues that are more likely to progress through the H3D drug discovery pipeline, accelerating the discovery of promising compounds across chemical series and disease areas.

## Discussion

We have introduced ZairaChem, a robust and parameter-free AutoML tool that makes use of a range of chemical descriptors in combination with an ensemble of AI/ML algorithms to train QSAR/QSPR models with SOTA performance out-of-the-box. Automation is key to ensure continuous integration and deployment of the AI/ML assets in an environment such as H3D, where data science capacity is (as yet) limited. We applied ZairaChem systematically to the existing drug discovery pipeline at H3D, yielding 15 production-ready models corresponding to key assays related to antimalarial and antitubercular screening cascades. ZairaChem models, whether trained on in-house data or from public data sources, showed excellent performance, with most AUROC scores above 0.75. The resulting hit enrichment reduces attrition rates of the experimental pipeline, potentially accelerating the bench-to-bedside turnaround time. For example, here we show that by testing just 50 compounds of the whole-cell screening data, we find four times more active compounds against *Pf* (NF54 strain) and *Mtb* (H37Rv strain) compared to testing the same number of compounds selected at random. This is particularly relevant in the context of research conducted within LMICs, where resources are typically constrained.

Subsequent analysis of local (series-specific) regions of the chemical space explored how model performance for specific chemical series is impacted by incrementally adding training compounds from the same chemical series. With limited local training data, model quality varies significantly from series to series; however, having approximately 30 local molecules within a chemical series appears to be a useful rule-of-thumb to produce models that can prioritise compound designs for further rounds of synthesis. Additional data, whether local or global, generally further enhances model performance.

Finally, we have demonstrated how AI/ML-aided decision-making can reduce attrition rates and accelerate project progression during de novo library screening by recreating a study performed at H3D in 2020. The visual representation of compounds' colour-vectors, based on AI/ML model scores, allowing for the quick identification and selection of the best compounds (highly potent against *Pf* with good solubility and low toxicity) for experimental testing. We further extended our model validation to the available prospective data in the year following the initial model training. The models showed good capacity for prioritising novel compounds from active chemical series across two different disease areas, expediting the discovery of promising drug leads with reduced resource expenditure. In summary, the ZairaChem virtual screening cascade will allow scientists to interrogate a broader chemical space and better prioritise compounds for synthesis and testing by taking into account predicted assay outcomes that might typically only be considered later in a drug discovery programme.

ZairaChem is a highly modular pipeline that, contrary to traditional 'static' methods, can be extended to cover specific tasks. The version described here is limited to single-output binary classification

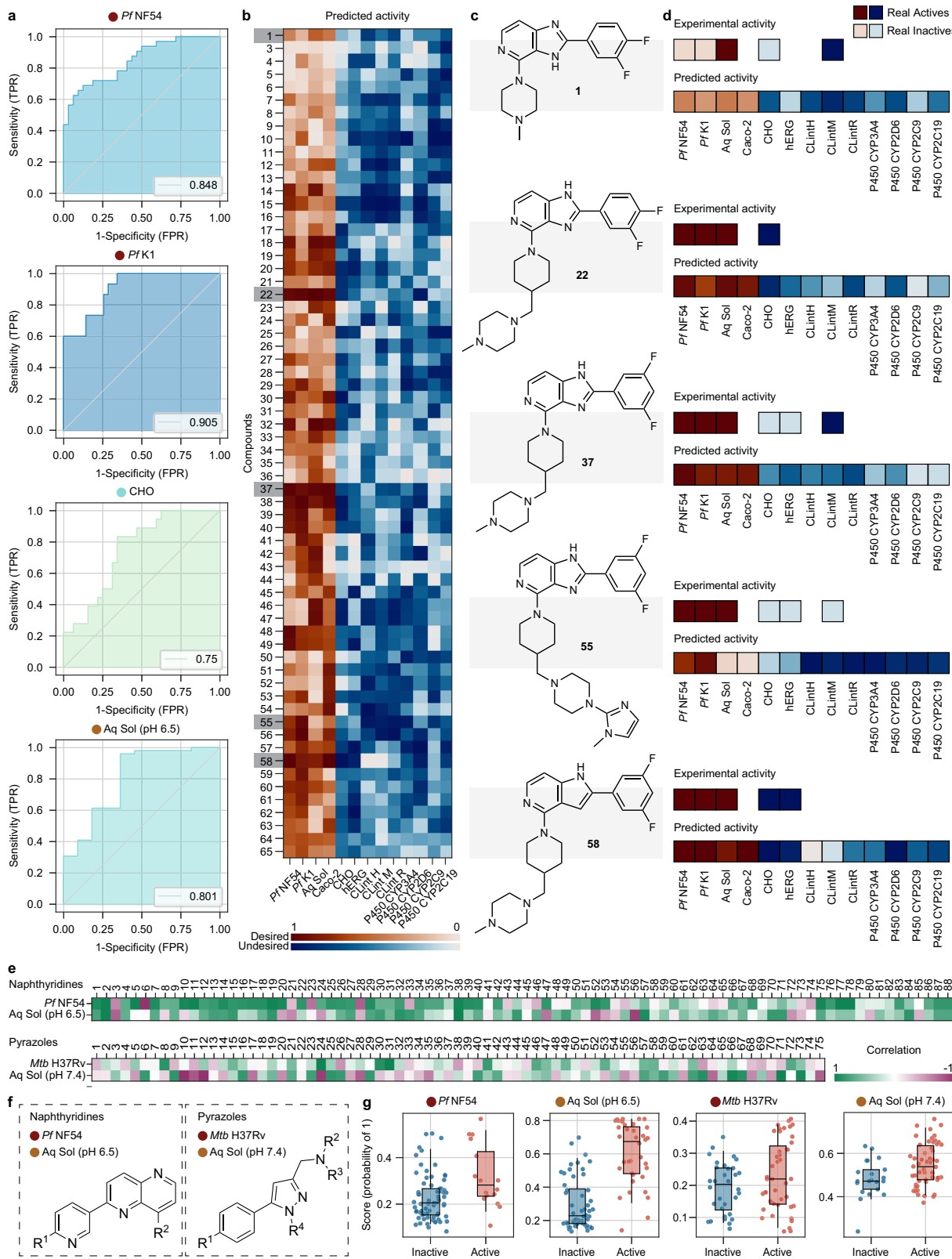

tasks, but the framework is prepared to deal with regression (i.e., continuous data) tasks. Likewise, although ZairaChem is now being extended in new directions, including interpretability, data augmentation for low-data regimes, and confidence estimation, these functionalities were not implemented in the current study, which is primarily focused on overall assessment of hit rates and predictive performance. Finally, ZairaChem can natively incorporate any AI/ML

model available from the Ersilia Model Hub[29], including modern descriptors, embeddings extracted from pre-trained models such as GEM (geometry-enhanced molecular representation)[32], or models fitted to a broad range of bioactivities using, for instance, the FS-Mol training set[33]. Here, we chose four representative descriptors corresponding to physicochemical properties, classical 2D structure fingerprints, advanced graph embeddings, and bioactivity signatures.

**Fig. 4 | De novo screening of libraries using AI/ML models.** Upper panel: **a** ROC curves of ZairaChem models tested on the library of 65 compounds (not included in the training set). Legend indicates the AUROC values of each model. Only models for which experimental validation was available for the 65 molecules are shown. **b** Predicted scores for each compound, transformed to a scale of 0 to 1 for comparison between assays. Desired activities are shown in a red colour scale and undesired activities are shown in a blue colour scale. Colour maps fade from 1 to 0 according to each model score. **c** Structure of selected compounds, including the initial hit compound 1. **d** Comparison of the predicted score and the experimental activity of selected compounds (non-existing squares indicate no experimental data on these assays). Experimental activity is represented as 1 (dark blue or dark red) or 0 (light blue, light red) for desired and undesired assay outcomes, respectively. Lower panel: Prospective validation for two active chemical series at H3D; naphthyridines active against *Pf* and pyrazoles targeting *Mtb*. **e** Model performance is depicted through correlations of model predictions with experimental results in which a green cell represents a correct model prediction while purple cells indicate incorrect predictions. **f** The core scaffold for each series is depicted as well as **g** a swarm plot for individual compound predictions. n active/inactive: *Pf* NF54 16/72, Aq Sol pH6.5 36/52, *Mtb* H37Rv 43/32, Aq Sol pH7.4 54/21. Boxes indicate the median (central line), Q1 (upper bound) and Q3 (lower bound) and whiskers extend to the data points up to 1.5 times in the interquartile range. Source data are provided as a Source Data file.

---

Improvement or extension of this default set is outside the scope of the study; however, this could serve to further improve the performance of the models presented here. It is worth noting that, in addition, the Ersilia Model Hub contains a growing number of predictive models for specific assay outcomes, developed either by third parties or by EOSI using publicly available data (e.g., from ChEMBL[34] or PubChem BioAssay[27]). For example, the MAIP antimalarial model[23] is readily accessible, as well as popular ADMET servers (SwissADME, ADMETlab 2.0)[35,36] and hERG inhibition models. ZairaChem can treat the outcome of these models as additional features for prediction. The version of the MAIP model in the Ersilia Model Hub, for instance, can predict H3D *Pf* NF54 outcomes with AUROC = 0.73 as a standalone estimator. Thus, the ZairaChem framework offers a unique opportunity to transfer knowledge from the public domain to in-house modelling efforts.

In conclusion, we have successfully developed and deployed an AI/ML-based QSAR/QSPR virtual screening pipeline based completely on open-source code and conventional computing capacity. All ZairaChem pre-trained H3D models are available for download and light versions can be explored in a web-based GUI (https://bit.ly/h3d-app). This is the first instance of a virtual screening cascade built with data produced on and for the African continent. We hope that the work presented here serves as a proof-of-concept for the potential of AI/ML tools to support drug discovery efforts in LMICs. Incipient centres in West Africa[37] and Central Africa[38] may benefit from similar implementations. More globally, ZairaChem offers a competitive, free, and constantly-updated software solution to model small-molecule bioactivity data, for which no strong data science skills are required to run the tool.

## Methods
### Data collection
Bioactivity data for the assays defined in Supplementary Table 1 were extracted from H3D's curated database that contains experimental data from the inception of H3D in 2010 up to November 2021. Each dataset was curated from a single set of assay conditions to minimise noise in the biological endpoints. During data pre-processing, compounds with large variations in replicated assay measurements were identified by first calculating the mean of the differences for all pairs of experimental values and then comparing this as a ratio to the overall mean experimental value to obtain a relative error. Compounds with a relative error greater than 1 (i.e., compounds with highly variable data compared to the mean) were removed from the dataset. The replicate values for remaining compounds were averaged regardless of date and site of experiment development. H3D's chemical and assay data was collected with the Dotmatics software.

Cytochrome inhibition data was obtained for CYP3A4, CYP2C9, CYP2C19, and CYP2D6 from PubChem (AID1851, AID899, AID891, AID883, AID884) and ChEMBL. PubChem BioAssay data were already binarized (1, active; 0, inactive). The ChEMBL database was queried and only compounds with "Standard Type", "IC$_{50}$" or "$K_i$", and "Standard Units": "nM" were considered. Compounds were assigned as "active" (1) if bioactivity was less than or equal to 10 μM, and "inactive" (0) otherwise. In addition, compounds with "Comment" equal to "Not Active" or "No Inhibition" were classified as inactive. Bioactivity data not matching these criteria were discarded.

### The ZairaChem pipeline
Each H3D assay was modelled independently using ZairaChem with default parameters. ZairaChem has two running modes, namely "fit" and "predict". ZairaChem training runs are scheduled to be executed twice a year for all assays in the virtual screening cascade. Extensive information on ZairaChem, especially for programmatic usage, is available online as part of the code repository and Ersilia's documentation pages.

**Data pre-processing.** The data pre-processing module consists of several steps. First, an input file is analysed and relevant columns are identified. In particular, the column containing SMILES strings is kept, together with the outcome (e.g., activity) column. ZairaChem determines the type of task (i.e., regression or binary classification). In this study, only binary classification tasks were considered. Small-molecule SMILES are standardised following the MELLODDY-Tuner protocol[39]. MELLODDY-Tuner is also used to identify LSH-based as well as Murcko scaffold-based splits, which can be used optionally. In addition, ZairaChem enables random and time-based splits, as well as a splitting scheme that takes into account clusters and LSH hashes to identify equally sized splits. By default, random stratified splits are done.

**Small-molecule descriptors.** ZairaChem can query the Ersilia Model Hub, EOSI's repository of pre-trained, ready-to-use AI/ML models. Some of the assets available in the Ersilia Model Hub provide as output a numerical vector for each molecule, typically capturing physicochemical or topological characteristics of the compound. By default, ZairaChem calculates such vectors for each molecule, corresponding to descriptor types that are representative of current approaches to small molecule featurisation. We expect this comprehensive strategy to favour good performance across a broad set of tasks, ranging from solubility prediction to whole-cell assays. In particular, we calculate (1) Mordred[14] descriptors (an array of >1600 physicochemical parameters), (2) ECFP fingerprints (a count-based vector of 2048 dimensions based on circular exploration (radius 3) of all atoms in a molecule), (3) Chemical Checker[16,40] signatures (a dense vector of 3200 dimensions capturing known and inferred bioactivity data across a wide range of bioactivity outcomes), (4) GROVER[41] embeddings (a 5000-dimension graph-based representation of the molecules), and (5) ChemGPT[42] embeddings (a chemical language model). Generally, we found the selected set of by-default descriptors to be a reasonable choice, with none of them being consistently better, or worse, than the rest across tasks. Other vectorial descriptors are available in the Ersilia Model Hub and can be specified to ZairaChem by simply referring to their model identifier. Of note, we also include the Ersilia Compound Embedding (https://github.com/ersilia-os/compound-embedding), a dense 1024-dimensional embedding based on an adaptation of the

prototypical network presented by FS-Mol, and pre-trained on ChEMBL bioactivity data. Details on the architecture and training procedure of the Ersilia Compound Embedding can be found in the corresponding code repository.

Continuous data descriptors are quantile-normalised and missing data is imputed with a nearest-neighbour approach. Invariant columns are removed. GROVER is used as a reference descriptor for additional processing. We perform PCA (four components) and UMAP (two components). In addition, supervised versions of these techniques are applied based on the binary outcomes. We choose linear optimal low-rank projection (lolP)[43] as an alternative to PCA for this supervised task. UMAP accepts both unsupervised and supervised modes. In addition to vector-like descriptors, it is possible to incorporate other models from the Ersilia Model Hub as auxiliary predictor variables for ZairaChem. ZairaChem treats these auxiliary models as additional columns in the pooling step described below.

**AutoML methods.** Currently, ZairaChem executes five AutoML methods independently. Each of the AutoML models is focused on enhancing a specific feature (e.g., interpretability, robustness, etc.) in the overall ZairaChem pipeline. Overall, the pipeline combines deep learning methodologies with tree-based methods and other classical ML approaches.

The first AutoML module performs independent modelling for each of the descriptors. The focus of this module is to identify which descriptor types are the most appropriate for the task of interest. With default parameters, five models are built, corresponding to the descriptors mentioned above. FLAML[18] is used to perform rapid search of Random Forest hyperparameters.

The second module is focused on visual interpretation of the chemical space. Thus, this module takes as input the low-dimensional PCA and UMAP projections obtained for the reference descriptor (in total, 12 variables). We use AutoGluon-Tabular[19] with default parameters to obtain robust classifiers and regressors.

The third module leverages GROVER, a data-driven descriptor trained on a large collection of molecules. This module is illustrative of a "transfer learning" approach where a large chunk of a neural network is "frozen" (the GROVER part) and a few extra layers are fine-tuned for the task of interest. Here, we add an additional dense layer. The number of dimensions of this layer, in addition to the training parameters, are automatically selected with Keras Tuner.

The fourth module leverages image-based representations of the molecules, enabling application of computer vision techniques, which are particularly advanced in the field of AI/ML. Compounds are represented as MolMaps[44]. These are concise, multi-descriptor images (maps) where regions of the image correspond to descriptors that are correlated. For example, in a MolMap one can find a region that relates to size and molecular weight, whereas other regions are related to lipophilicity, solubility, etc. It has been shown that convolutional neural networks can be used out-of-the-box taking MolMaps as input, without the need for intense architecture and hyperparameter search.

Finally, the fifth module includes the TabPFN classifier, a novel, fully-trained transformer network that is able to perform Bayesian inference in a single forward pass[21]. TabPFN is limited to small tabular classification tasks of up to 1000 samples and 100 dimensions. We used the TabPFN with the aforementioned descriptors, reducing their dimensionality from 1024 to 100 using lolP. Since many datasets contain more than 1000 samples, we devised a subsampling strategy based on three imbalanced learning strategies (imb-learn package), namely K-means SMOTE (for oversampling), edited nearest neighbours (for undersampling) and a combination of over and undersampling with SMOTE-Tomek.

ZairaChem is prepared to be extended with additional AutoML modules, if and as necessary. However, the trade-off between computing time and gain in performance is an important consideration before adding further modules.

**Pooling.** Each of the models above provides point predictions that can be aggregated in a consensus (pooled) prediction. By default, ZairaChem applies a "blending" approach, based on a weighted average between individual predictions. The default weighting scheme is based on the estimated performance of the individual predictors. Prior to aggregation, prediction scores are power- and logit-transformed.

**Reports and output.** At the end of the ZairaChem pipeline, performance reports are automatically provided, including the most common validation metrics for binary classification tasks. A single spreadsheet with prediction output and performance metrics is provided as a primary result, along with multiple reporting plots (AUROC, PCA, UMAP projections, etc.). We use both PCA and UMAP dimensionality reductions as complementary methods to visualise the chemical space: a PCA plot is a linear rescaling of the data that provides a global overview of the dataset, while a UMAP plot is a nonlinear rescaling method that emphasises the local clustering of structurally-related compounds and provides insight into data homogeneity.

**Web-based predictions.** Interaction with light versions of the H3D models is possible through a GUI in a web application (https://bit.ly/h3d-app). Light models were created using FLAML on Ersilia Compound Embeddings, based on the observation that, inside the ZairaChem ensembles, both components have good performance across H3D prediction tasks. Light models preserve >95% of the original performance (Supplementary Table 6). In the prediction app, we provide the classification score along with a percentile calculated with respect to a representative set of ca. 200,000 molecules extracted from ChEMBL.

In addition, we provide fully equipped versions of the ZairaChem models for local runs. All IP-sensitive small molecule structures have been removed from any ZairaChem model folder.

**ZairaChem benchmarking.** We selected Therapeutics Data Commons (TDC) as a benchmark framework. TDC contains multiple datasets across a broad range of tasks and activities. To prove SOTA performance on H3D's data, we selected the TDC ADMET Group as the one containing tasks most similar to the ones described in this project. In line with this, only classification tasks were evaluated. Data were downloaded from TDC and split into train and test according to the TDC guidelines. Models were trained with default ZairaChem parameters with five-fold cross-validation. Results were calculated using TDC guidelines.

### Analysis of H3D's chemical series

To investigate the change in model performance within a localised region of chemical space through the incremental addition of local training data, chemical series with at least 200 compounds were selected from the H3D library for analysis.

**Train-test splits.** Datasets for model training were constructed according to the following protocol: first, all compounds in a series of interest were removed from the bulk global training data of the H3D library; then, the data was shuffled and 100 compounds were randomly selected with stratification as a standard test set; the remaining 100 molecules were systematically added back to the bulk H3D activity data and models trained for each dataset. A final separate 'local-only' model was trained on the 100 series-specific compounds alone, without the broader H3D library data, in order to investigate the effect of global training data on model performance.

**AUROC percentage change.** To measure the contribution of additional global training data to a model's performance in a localised

chemical space, we calculate the percentage change in AUROC score according to the following steps: (1) first, we find the difference between the AUROC scores of the model trained on 100 series-specific compounds with the H3D library included as well as the 'local-only' model trained on the series-specific compounds alone; (2) next, for the 'local-only' model, we find the AUROC score that is still possible to be achieved (1 - 'local-only' AUROC); (3) lastly, we take the difference in model scores (from step 1) as a percentage of the score that could still be achieved (from step 2). This metric represents the additional performance gained or lost out of the total AUROC score that was still available to capture. AUROC scores were calculated with the SciKit-Learn package from a five-fold cross-validation.

**Plots.** Chemical space visualisation was constructed by describing compounds using the Morgan Fingerprint algorithm, with a radius of 3 and vector length of 2048 as implemented in the RDKit package. These fingerprints were then projected onto two dimensions through principal component analysis as well as the UMAP algorithm and plotted with the Matplotlib library.

### Cell lines for experimental screening
CHO and HepG2 mammalian cell lines for cytotoxicity assays, the Caco-2 cell line for permeability studies, and the H37Rv strain of *Mycobacterium tuberculosis* were all acquired from ATCC (American Type Culture Collection). Strains of *Plasmodium falciparum* (NF54 and K1) came from the Malaria Research and Reference Reagent Resource Center (MR4) at the ATCC, now called BEI Resources.

### Inclusion and ethics statement
The research reported in this shared-authorship publication results from a joint partnership during an on-site research visit of the Ersilia Open Source Initiative to the H3D Centre in Cape Town. Roles and responsibilities were agreed to when the project commenced and with continuous communication throughout. The majority of AI/ML models were trained on in-house data to accelerate drug discovery research efforts at the H3D Centre, where no prior local AI/ML infrastructure or skills existed. This project, together with various seminars and workshops, facilitated the development of local AI/ML capacity that could be intentionally sustained after the conclusion of the research visit by EOSI. In addition, these skills are intended for further dissemination to researchers based in Africa through the H3D Centre's capacity-building initiatives. While drug discovery in Africa is in a nascent state, we have endeavoured, where possible, to reference relevant publications and initiatives on the African continent, e.g., refs. 37,38, respectively.

### Reporting summary
Further information on research design is available in the Nature Portfolio Reporting Summary linked to this article.

## Data availability
This work contains AI/ML models built on top of H3D proprietary data. Data are managed at H3D with the Dotmatics software. The majority of the compounds used for training are proprietary data from the H3D Centre from active drug discovery research programmes which are subject to the intellectual property management plans as per funding and collaboration agreements. Access requests to the raw model training data for the *Pf* NF54, *Pf* K1, *Mtb* H37Rv, CHO, HepG2, Clint H, M and R, Caco-2 and Aq Sol can be made to susan.winks@uct.ac.za with an expected response within two days. However, all corresponding AI/ML models are publicly available. ChEMBL and PubChem BioAssay were used as additional sources of data for the CYP450 models. Raw data can be directly accessed on the PubChem and ChEMBL websites (https://pubchem.ncbi.nlm.nih.gov; https://ebi.ac.uk/chembl) and processed data can be found here: https://bit.ly/h3d-ext-data. Benchmarking of ZairaChem was performed using the Therapeutics Data Commons ADMET classification tasks, available at https://tdcommons.ai. Source data are provided in this paper.

## Code availability
ZairaChem code is available at https://github.com/ersilia-os/zaira-chem. Extended documentation can be found in the Ersilia Book (https://ersilia.gitbook.io). ZairaChem benchmarks are reported at https://bit.ly/h3d-tdc-bench. Code used for analysing data can be found at https://bit.ly/h3d-code. Download links to the fully equipped ZairaChem models are available at https://bit.ly/h3d-models. A light version of the H3D models is available as a web-based app at https://bit.ly/h3d-app. Code for deployment can be found at https://bit.ly/h3d-app-code.

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

## Acknowledgements

EOSI is grateful to Merck KGaA for a Biopharma Speed Grant. J.H. is a recipient of the Harry Crossley Foundation Postdoctoral Fellowship. Capacity-building activities within the context of the H3D-EOSI collaboration were supported by an Event Fund grant from Code for Science & Society and the Wellcome Trust; we thank Dr Susan Winks for her support in coordinating this event. K.C. is the Neville Isdell Chair in African-centric Drug Discovery and Development and thanks Neville Isdell for generously funding the Chair. The South African Medical Research Council and South African Research Chairs Initiative of the Department of Science and Innovation are gratefully acknowledged for their support (K.C.). The authors would like to thank Dr Jake M. Pry for providing feedback on the manuscript, Dr André Horatscheck and Dr Grant Boyle for kindly facilitating access to the central database at H3D, as well as Dr Preshendren Govender for sharing the *M. tuberculosis* screening data.

## Author contributions

M.D.F., G.T., and K.C. designed the study. M.D.F. developed ZairaChem with the support of J.H., G.T., and A.K. G.T. performed the benchmarking. G.T. curated and analysed the data with the help of J.W. and J.H. M.D.F., G.T., and J.H. performed the analysis. J.H. and J.W. implement and maintain the models at H3D. K.C. and M.D.F. supervised the project. All authors discussed the results and commented on the manuscript.

## Competing interests

The authors declare no competing interests.
