## [Peer Review File · Nature Communications]

First fully-automated AI/ML virtual screening cascade implemented at a drug discovery centre in AfricaREVIEWER COMMENTS

Reviewer #1 (Remarks to the Author):

The paper "First fully-automated AI/ML virtual screening cascade implemented at a drug discovery centre in Africa" by Turon et al describes the development of ZairaChem an automated AI/ML platform for training QSAR models with potential application for drug discovery, with a potential emphasis on malaria and mtb research. The platform incorporates 15 models created on in-house and literature datasets and demonstrated adequate performance with various AUROC parameters in >0.7 range, when individually benchmarked against the state of the art tools.

While the developed binary models are not particularly novel, and rely on established QSAR descriptors and algorithms, the idea of developing an open source platform for researchers from African continent is far-reaching and deserving. Authors also retrospectively simulated the discovery of a series of 2,4-disubstituted imidazopyridines as antiplasmodial agents and demonstrated that ZairaChem could qualify them with AROC=0.74. This example is important.

However, while the paper is technically sound and the goal is noble, authors might need to consider adding more 'punchline' materials in order to make it attractive for the Nature series readership (while this paper would certainly already be welcomed by journals like JCIIM). Thus, this work needs some original discoveries, such as identification and validation of novel, previously untested agents active against mtb or plasmodium. The corresponding experimental section would significantly strengthen the paper and would add needed impact.

Authors should also provide more user guidance, as the platform is intended for public use and relies on the open source concept. Ideally, some simple to use web implementation with user-friendly GUI should be provided in order to fully capture the value for the proposed platform.

But overall, this is an important initiative that will be well received by the cheminformatics community.

Reviewer #2 (Remarks to the Author):

The paper of Turon et al describes the development of AI/ML for prediction of whole-cell growth inhibition assays for malaria and tuberculosis as well as for some pharmacokinetic and toxicity properties, mainly using proprietary datasets from H3D center, but also using some public available datasets. The paper is very well written and easy to follow. In my opinion, the paper deserves publication, but I have some comments and issues that should be taken in consideration before publication.

In the field of drug discovery and the computer-aided drug discovery, this task is known as Quantitative Structure Activity/Property Relationship Modeling (QSAR/QSPR). So, first, authors should avoid using fashion names such as AI/ML and use the correct name of what they are using / developing.

Also, since the beginning of the paper, it should be stated that the developed models and the pipeline are available for the community. This is not clear in the text.

I did not find the datasets in the git. Are they available?

Also, authors should compare their models and pipeline with the the established literature, for instance, there are many models for hERG, for CYP, for cytotoxicity, and also for Mtb and Pf. Also, they should compare with other pipelines.

Unfortunately, the manuscript has a very limited impact on other researchers, as the datasets used for the models are very limited, some have around 100 compounds, which is considered low data nowadays. Where data is scarce, authors should have tried to use transfer learning or few-shot learning.

In my opinion, the major flaw of this work is that the authors did not test their pipeline for a “real-world” example, which is a bit disappointing. They have only demonstrated the effectiveness of the cascade by trying to reproduce an already described discovery of antimalarial compounds. So, in my opinion, to worth the publication in Nat Comms, authors should really validate their pipeline by testing and proving that their models are really able to accelerate the drug discovery process.^[1]

The abstract is not very accurate and informative - for example: in the abstract is stated that the datasets used involve drug metabolism properties. However, the manuscript used besides the Pf and Mtb datasets, cytotoxicity data in two cell lines (CHO, HepG2) and some pharmacokinetics properties such as aqueous solubility (Aq. sol.), Caco-2 permeability (Caco-2) and intrinsic clearance (CLint) for human (H), mouse (M), and rat (R) microsomes., CYP inhibition and hERG cardiotoxicity. Therefore, the Abstract should be fully revised to give a better overview of the work.

Did the authors tried to use some DL architheture? Why not?

Page 03 - line 78 - Authors start the results with figure 2a. But Figure 1 has not been cited in the text before this.

Authors built only classification models. It would be beneficial for the community to have also some regression models.

Page 04, line 130. When data points were scarce, why didn't the authors tried transfer learning or few shot learning?

* 10% of the data was held as a test set. This seems to me too little compounds in the test set. Usually, it is kept with 20% of the whole data set.

* Moreover, did the authors used an external test set?

* Datasets were very unbalanced (eg. ~15% actives to 85% of inactive). How did the authors deal with this? Did the authors tried to balance the datasets?

* How were the activity cut-offs selected? Are there any rational why for Pf it was 0.1 uM and for Mtb it was 5 uM?

* How were the descriptors chosen?

* Compounds seem to have many chiral centers. How did the authors deal with the stereochemistry of the compounds? Did the activity or biological endpoint was measured on the pure stereoisomers or racemic mixture?

* Why did the authors chose to use both PCA and UMAP for analyzing the chemical space? this should be better explained and discussed.

* Methods section: In my opinion, it is very general and brief and impossible to reproduce the work.

* Authors should make all files, datasets and scripts available in a git hub so we could test the pipeline and all the community could benefit from this work.

* how were the compounds curated? And how about the duplicates were analyzed? In case that the same compound was tested on two or more different assays, how was this analyzed and deal?

* For the benchmarking, other datasets should be used. ^[1-3]_[SEP]

Response to Reviewers' Comments

Reviewer #1

The paper "First fully-automated AI/ML virtual screening cascade implemented at a drug discovery centre in Africa" by Turon et al describes the development of ZairaChem an automated AI/ML platform for training QSAR models with potential application for drug discovery, with a potential emphasis on malaria and mtb research. The platform incorporates 15 models created on in-house and literature datasets and demonstrated adequate performance with various AUROC parameters in >0.7 range, when individually benchmarked against the state of the art tools.

While the developed binary models are not particularly novel, and rely on established QSAR descriptors and algorithms, the idea of developing an open source platform for researchers from African continent is far-reaching and deserving. Authors also retrospectively simulated the discovery of a series of 2,4-disubstituted imidazopyridines as antiplasmodial agents and demonstrated that ZairaChem could qualify them with AUROC=0.74. This example is important.

Response: We would like to thank the Reviewer for the positive and encouraging remarks, in particular highlighting (a) the adequate performance of our AI/ML pipeline, (b) the scope of our work and mission with respect to drug discovery in Africa, and (c) the importance of the experimental validation example.

As suggested by the Reviewer, we have extended our study with additional examples.

However, while the paper is technically sound and the goal is noble, authors might need to consider adding more 'punchline' materials in order to make it attractive for the Nature series readership (while this paper would certainly already be welcomed by journals like JCIM). Thus, this work needs some original discoveries, such as identification and validation of novel, previously untested agents active against mtb or plasmodium. The corresponding experimental section would significantly strengthen the paper and would add needed impact.

Response: We thank the Reviewer for bringing this to our attention. We acknowledge and agree with the Reviewer that documenting prospective, original discoveries would significantly strengthen this paper. Hence, we have accordingly now included in the revised manuscript two prospective studies on active projects at the H3D Centre, and are pleased to reveal the general chemical structures for these chemotypes representing novel, previously-untested agents in both the malaria (*P. falciparum*) and tuberculosis (*M. tuberculosis*) disease areas. While detailed experimental values and protocols will be documented in separate publications, we are able to share how our AI/ML tools were validated with previously unseen compounds and how the agreement between predicted and experimentally-obtained values was favourable. We believe that, collectively, this provides strong evidence of a 'real-world' application of the presented AI/ML toolbox.

In addition to a new paragraph in the text, which we have highlighted in the revised manuscript, we have expanded Figure 4 to include the prospective application of the virtual screening cascade models to novel analogues from two active medicinal chemistry programmes at H3D. We show the model performance for assays with sufficient experimental data to provide

meaningful model validation; specifically, whole-cell activity (*Pf* NF54 and *Mtb*, respectively) and solubility. Our models show satisfactory performance for these previously untested compounds in terms of precision (P) and recall (R) at stringent (s) and more permissive (p) cutoffs. For the so-called naphthyridine series: *Pf* NF54 Ps = 0.667, Rs = 0.118, Pp = 0.333, Rp = 0.529; Solubility Ps = 0.765, Rs = 0.703, Pp = 0.648, Rp = 0.946. For the pyrazole series: *Mtb* Ps = 0.778, Rs = 0.298, Pp = 0.577, Rp = 0.872; Solubility Ps = 0.814, Rs = 0.625, Pp = 0.727, Rp = 1.0. Swarm plots showing the difference in scores between active and inactive compounds can be found in the updated Figure 4. This ‘real-world’ example illustrates the capacity of the ZairaChem pipeline to produce AI/ML models that significantly accelerate the drug discovery process in a resource-constrained setting by prioritising prospective compounds for synthesis that are much more likely to progress through the drug discovery pipeline.

Authors should also provide more user guidance, as the platform is intended for public use and relies on the open source concept. Ideally, some simple to use web implementation with user-friendly GUI should be provided in order to fully capture the value for the proposed platform.

Response: We thank the Reviewer for acknowledging the open source model we adhere to throughout the paper. Accessibility and reproducibility are especially essential in the context of low-resource settings, and we agree with the Reviewer that an easy-to-use interface would benefit researchers outside H3D and increase the usability of the models. To that end, we have deployed a light version of the models through a graphical user interface (GUI). We invite the Reviewer to try it out on: h3dscreening.ersilia.io [link].

Achieving this deployment demanded substantial additional work, which is now included in the manuscript. In brief, after developing the Ersilia Compound Embedding, a data-driven embedding descriptor optimised for transfer and few-shot learning (<https://github.com/ersilia-os/compound-embedding>), we have trained lightweight versions of the H3D models based on the tabular FLAML AutoML framework. We have observed that, systematically, these light models retain >95% of the performance of the fully-equipped ones (Extended Data Table 6). We have also developed a simple StreamLit app (<https://github.com/ersilia-os/h3d-screening-cascade-app>) as a backend to serve model predictions through the web app interface from which users can submit their queries as a list of SMILES strings. In addition, the models have been incorporated in the Ersilia Model Hub to enhance discoverability and so that they can be used together with other models in the Hub.

Importantly, full versions of the models are now also available for download: <https://github.com/ersilia-os/h3d-screening-cascade-models>. In this case, to respect H3D IP protection over training set compounds, models have been “anonymised” to avoid chemical structure disclosure. We expect the addition of the “Anonymisation” feature in the ZairaChem pipeline will also enhance the use of the AutoML tool by other researchers concerned with the privacy of their training sets, as is the case for H3D.

Moreover, we have substantially extended the user CLI documentation in the README file of the repository and the official ZairaChem documentation. Both documents are cited in the manuscript.

Overall, we are grateful to this Reviewer for raising this valid point. We feel that these necessary improvements to usability have notably expanded the reach of our work.

But overall, this is an important initiative that will be well received by the cheminformatics community.

Response: We thank the Reviewer for their positive feedback and for highlighting the impact of our work on the cheminformatics community. We would also like to emphasise that the end-to-end implementation in a Global South institution in Africa demonstrates that AI/ML tools can be developed and implemented without extensive infrastructure. This should encourage researchers outside the cheminformatics community to increase the adoption of the technology in their projects.

Reviewer #2

The paper of Turon et al describes the development of AI/ML or prediction of whole-cell growth inhibition assays for malaria and tuberculosis as well as for some pharmacokinetic and toxicity properties, mainly using proprietary datasets from H3D center, but also using some public available datasets. The paper is very well written and easy to follow. In my opinion, the paper deserves publication, but I have some comments and issues that should be taken into consideration before publication.

Response: We thank the Reviewer for their appreciation of the manuscript and the work presented. We are grateful to the Reviewer for recommending the publication of our work, and the Reviewer's constructive feedback has contributed to the improvement in the overall quality of the manuscript. Though the publication describes a relatively complex computational framework, we have intentionally emphasised clarity and conciseness in the hope of making it accessible and interesting to a wider audience, with the end goal of increasing uptake of AI/ML tools by experimental researchers working in low-resource settings. We have carefully discussed the Reviewer's comments below, and reviewed the work to address their concerns.

In the field of drug discovery and computer-aided drug discovery, this task is known as Quantitative Structure Activity/Property Relationship Modeling (QSAR/QSPR). So, first, authors should avoid using fashion names such as AI/ML and use the correct name of what they are using / developing.

Response: We agree with the Reviewer that the synonymia between QSAR/QSPR and AI/ML was not clear in the original manuscript. We now make it more clear throughout the revised manuscript with revisions highlighted in yellow. We acknowledge the Reviewer's comment that, indeed, molecular activity/property prediction based primarily on chemical structure (as we do here) falls within the QSAR/QSPR approach. ZairaChem is a flexible, extensible framework where new descriptors can be included, as well as new supervised learning tools (classifiers, in the current case). Some of the by-default descriptors used are deep learning embeddings, like GROVER, ChemGPT or Chemical Checker descriptors, and others can be easily included from the Ersilia Model Hub catalogue, such as MolBERT. As for the classifiers,

we use a set of AutoML techniques, both via a “classical” set of tools (e.g. XGBoost in FLAML and AutoGluon) as well as neural network exploration via e.g. Keras Tuner. The MolMapNet convolutional network is also used.

To address the Reviewer’s concern, we have made two substantial additions to ZairaChem, both of them falling within the state-of-the-art of AI/ML. First, we have developed a novel compound descriptor, the Ersilia Compound Embedding (<https://github.com/ersilia-os/compound-embedding>), focused on transfer and few-shot learning. In brief, this embedding descriptor follows the FS-Mol approach based on ChEMBL data (<https://github.com/microsoft/FS-Mol>), with modifications to capture GROVER and Mordred descriptors which, in our hands, are highly performant in many of the classification tasks. The resulting Ersilia Compound Embedding has 1,024 dimensions and has been successfully incorporated into the ZairaChem pipeline in a distilled (lightweight) form. Second, we have incorporated a genuinely new classifier to the ensemble of AutoML tools. In particular, the ZairaChem ensemble now uses TabPFN (<https://github.com/automl/tabPFN>), a fully-trained transformer network for tabular classification that performs Bayesian inference with a single forward pass. To the best of our knowledge, the recently published TabPFN (Hollman, 2022) has not been previously used in QSAR/QSPR.

Collectively, the ZairaChem pipeline thus involves a significant number of AI/ML methodologies. We have included mentions to the new AI/ML methods with hopes that a better and more appropriate use of the QSAR/QSPR and AI/ML terms in the text will be found.

Also, since the beginning of the paper, it should be stated that the developed models and the pipeline are available for the community. This is not clear in the text.

Response: We would like to thank the Reviewer for raising this valid point. In line with Ersilia’s mission of equipping laboratories in the Global South with AI/ML tools for infectious disease research, we have developed a light version of the models for deployment, and we have created an easy-to-use Graphical User Interface (GUI) to improve their usability and to encourage uptake at other institutions outside H3D. This can be accessed via the web-based version of the app: h3dscreening.ersilia.io [link]. Correspondingly, the code for producing this app can be found in this new GitHub repository: <https://github.com/ersilia-os/h3d-screening-cascade-app>. In addition, the H3D models have been included in the Ersilia Model Hub (<https://ersilia.io/model-hub>) and a downloadable version is available here: <https://github.com/ersilia-os/h3d-screening-cascade-models>. This has also been clarified in the revised version of the manuscript, and the links to the models and usage instructions can be found in the Code and Data Availability section.

In sum, in the manuscript revision process, we put significant emphasis on improving the accessibility of our H3D models.

I did not find the datasets in the git. Are they available?

Response: The majority of the models are developed with proprietary data and compound collections from the H3D Centre which, due to intellectual property (IP) constraints, cannot be publicly disclosed. In our opinion, these constraints advocate for the release of open-source AI/ML models, especially when they are built on IP-sensitive data, since these models can then act as a “surrogate” way of releasing the information accumulated at the H3D Centre over the last decade without threatening their IP position. Similar initiatives have been undertaken

by large pharmaceutical companies and consortia, and have been well received by the scientific community. For example, the MAIP model, a predictor of antimalarial activity, has been developed thanks to a public-private partnership between five pharmaceutical companies and non-profit partners, and the model is available online at EMBL-EBI, but the >7 M compounds used to train it has not been disclosed (Bosc et al, 2021). Another example would be the MELLODDY consortium, that centralises private bioactivity data from multiple stakeholders by means of AI/ML modelling. Along these lines, in the process of this revision, we have added an “anonymisation” flag in the ZairaChem pipeline that allows us to safely release the fully-trained models for the community. All models are now available for download, as mentioned in the previous comment.

The cytochrome P450 AI/ML models have been built using publicly-available data (ChEMBL and PubChem) on cytochrome inhibition, as described in the manuscript. Public curated data for the CYP P450 models is available here.

Also, authors should compare their models and pipeline with the established literature, for instance, there are many models for hERG, for CYP, for cytotoxicity, and also for Mtb and Pf. Also, they should compare with other pipelines.

Response: We agree with the Reviewer that benchmarking our AutoML pipeline (ZairaChem) against established datasets and tools is important to demonstrate its performance outside the H3D chemical space. Indeed, in Figure 2 we report two use-cases outside the H3D datasets. First, the CYP P450 AI/ML models have been built using public data from ChEMBL and PubChem, and then used to predict the activity of H3D compounds (Figure 2 and Extended Data Figure 2 and 3). Second, the hERG model used to predict hERG activity in the H3D dataset is a deep learning model developed by Karim et al, 2021, that demonstrates good predictive potential on the H3D chemical space (Figure 2). In addition, the ZairaChem pipeline was benchmarked with the Therapeutics Data Commons (TDC) set of 13 ADMET binary classification tasks, which include predictions for standardised datasets for: bioavailability, P-glycoprotein inhibition, hERG inhibition, CYP inhibition, blood-brain-barrier permeation, mutagenicity, and liver toxicity. The benchmarking results can be found in Extended Data Table 2, where out-of-the-box ZairaChem models scored **between 1st and 4th place on the TDC ADMET leaderboard in a five-fold cross-validation**. We have included additional text in the revised manuscript to make the importance of this benchmarking result more explicit.

Finally, as suggested by the Reviewer, we have further extended the benchmarking to include available open source models in the literature. To understand to what extent models developed with external data and different AI/ML methods can be applied to the H3D use-case, we have selected a version of the MAIP model available from the Ersilia Model Hub (<https://github.com/ersilia-os/eos2gth>) (Bosc et al, 2021) and the ChemTB model (Ye et al, 2021). These models predict the bioactivity against *P. falciparum* and *M. tuberculosis*, respectively, and are publicly available. We observe how they retain some predictive potential in the H3D dataset (AUROCs between 0.74 and 0.62) but they do not attain the same level of performance as the models developed with ZairaChem (*Pf* NF54 0.902, *Mtb* 0.903). For the ADMET properties, we have chosen the NCATS@ADME toolbox, which offers AI/ML models for P450 cytochrome inhibition (CYP3A4, CYP2C9, CYP2D6), aqueous solubility and human and rat metabolic clearance. Again, AUROCs for H3D datasets in the ADME@NCATS toolbox range between 0.76 and 0.62, yielding inferior performances to the ZairaChem models reported in Extended Data Table 3. This allows us to conclude that developing models with

in-house data using automated AI/ML pipelines adds significant value to modelling efforts and enables the selective identification of potential bioactive, drug-like compounds in the H3D collection. These results have been added in Extended Data Figure 1 and are referenced in the text.

Unfortunately, the manuscript has a very limited impact on other researchers, as the datasets used for the models are very limited, some have around 100 compounds, which is considered low data nowadays. Where data is scarce, authors should have tried to use transfer learning or few-shot learning.

Response: While the majority of the AI/ML models have been trained with datasets containing over a thousand compounds, we agree that some models, like the Caco-2 permeability model, are low data (Extended Data Table 1). In any case, it is true that the number of “actives” is limited in some datasets, which necessarily limits the domain of applicability of the tools. The Reviewer’s suggestion to include transfer and few-shot learning in our dataset is an excellent one and, indeed, this is the main driver behind the newly added Ersilia Compound Embedding. By design, this embedding captures bioactivity data from ChEMBL and, conjointly, uses pre-trained (transfer-learning) embeddings such as GROVER. This embedding procedure is now incorporated in the default ZairaChem configuration, along with another transfer learning approach (Chemical Checker, Duran-Frigola et al, 2020). In addition, as mentioned in an earlier response, we have also incorporated TabPFN, a classifier specifically developed to resolve small classification tasks.

In addition, in cases where in-house data are scarce, we have integrated data from external sources, like in the case of the hERG and CYP models. We leverage a combination of CYP data from public databases (~15 000 compounds) and 30 H3D compounds to improve CYP model performance in the H3D chemical space and we demonstrate the resulting improvement in model performance in Extended Data Figure 6. For the hERG use-case, we leverage CardioToxNet, a ready-made AI/ML model developed by Karim et al, 2021. In a broader sense, the Ersilia Model Hub (a resource plugged to ZairaChem) is an ever-growing repository of ready-to-use AI/ML models. As more models are incorporated in this resource, we envisage that they can be selectively used as auxiliary inputs for ZairaChem, thus providing yet another means to perform transfer learning.

Finally, to provide guidance for model users (e.g. medicinal chemists), we have investigated the contribution of general data versus specific chemical series data points to model performance by training models on datasets consisting of 100 series-specific compounds (Figure 3c, 3g). The data drop-out study shows how, while the addition of local training points improves model performance (as expected), the presence of ~30 compounds from a specific chemical series to the “global” dataset is often sufficient to provide good predictive performance. This exercise can provide a valuable guide for researchers who want to apply the ZairaChem pipeline to their own datasets.

In my opinion, the major flaw of this work is that the authors did not test their pipeline for a “real-world” example, which is a bit disappointing. They have only demonstrated the effectiveness of the cascade by trying to reproduce an already described discovery of antimalarial compounds. So, in my opinion, to worth the publication in Nat Comms, authors should really validate their pipeline by testing and proving that their models are really able to accelerate the drug discovery process.

Response: We thank the Reviewer for this feedback and agree with the valid point they have raised vis-à-vis testing the pipeline against a ‘real-world’ example. To address this point, we have included two prospective studies in the revised manuscript. We have included an additional supplementary figure (Extended Data Figure 7) showing the prospective performance of the in-house models, which were trained on data up to 2021, for data produced across all H3D Centre projects in 2022. Furthermore, we have expanded Figure 4 to include a prospective study for two specific chemical series, spanning both malaria (*P. falciparum*) and tuberculosis (*M. tuberculosis*) disease areas, for assays in which sufficient experimental data are available for model validation. These include *Pf* NF54 and *Mtb* predictions for the so-called naphthyridine and pyrazole chemical series, respectively, as well as aqueous solubility predictions.

As previously described in response to Reviewer 1, the model predictions show satisfactory prediction performance in terms of precision (P) and recall (R) at stringent (s) and more permissive (p) cutoffs. For the naphthyridine series: *Pf* NF54 Ps = 0.667, Rs = 0.118, Pp = 0.333, Rp = 0.529; Solubility Ps = 0.765, Rs = 0.703, Pp = 0.648, Rp = 0.946. For the pyrazole series: *Mtb* Ps = 0.778, Rs = 0.298, Pp = 0.577, Rp = 0.872; Solubility Ps = 0.814, Rs = 0.625, Pp = 0.727, Rp = 1.0. Figure 4 of the manuscript has been updated to show these results, including swarm plots to show the difference in scores between active and inactive compounds. The implication of these results is that by applying these models, medicinal chemists at H3D now have a computational aid to develop the structure-activity relationships much more efficiently and, therefore, more quickly identify promising compounds to advance through the drug discovery pipeline. As we describe in the revised version of the manuscript, these projects have yielded compounds with improved pharmacokinetic and efficacy profiles relative to those previously reported for this chemotype (in the case of the antimalarial series) and compounds with improved potency, aqueous solubility and cardiotoxicity margins (in the case of the antituberculosis series). Experimental details relating to the specific compounds are beyond the scope of this manuscript and will be described in separate publications.

The abstract is not very accurate and informative - for example: in the abstract it is stated that the datasets used involve drug metabolism properties. However, the manuscript used besides the *Pf* and *Mtb* datasets, cytotoxicity data in two cell lines (CHO, HepG2) and some pharmacokinetics properties such as aqueous solubility (Aq. sol.), Caco-2 permeability (Caco-2) and intrinsic clearance (CLint) for human (H), mouse (M), and rat (R) microsomes., CYP inhibition and hERG cardiotoxicity. Therefore, the Abstract should be fully revised to give a better overview of the work.

Response: We agree and thank the Reviewer for this comment. We have revised the text in the Abstract to better and more accurately reflect the focus of this work within the word limit; that is, stating explicitly the assays for which models have been developed. Edits to the revised manuscript are highlighted in yellow.

Did the authors try to use some DL architecture? Why not?

Response: Yes, indeed. ZairaChem used DL architectures and, as mentioned above, we have made new additions in this direction. Amongst the by-default DL architectures included are: GROVER (a graph-based transformer); ChemGPT; Chemical Checker “signaturizers”; TabPFN; KerasTuner MLPs; MolMapNet (convolutional), one of the AutoGluon components; and the Ersilia Compound Embedding, which capitalises on prototypical networks.

Page 03 - line 78 - Authors start the results with figure 2a. But Figure 1 has not been cited in the text before this.

Response: We thank the reviewer for pointing out this mistake. We have removed the reference to Figure 2a on the third page and have instead explicitly written out the whole-cell and biochemical ADMET properties to which we are referring.

Authors built only classification models. It would be beneficial for the community to also have some regression models.

Response: We agree with the Reviewer that regression tasks can be valuable to the community. In this study, we have focused on binary classifications since the goal is to speed up the decision-making process at the H3D Centre, which employs go/no-go decisions based on expert-determined experimental cut-offs. A sentence about selection of cut-offs through consultation with experts has been added in the text. Our next steps include the development of ZairaChem regression models, but this is outside the scope of the present manuscript.

Page 04, line 130. When data points were scarce, why didn't the authors try transfer learning or few shot learning?

Response: We have addressed this point in our response to the comment above with respect to datasets of 100 compounds, few-shot learning and transfer learning. We have improved the manuscript to include explicit mention of transfer learning and few-shot learning.

* 10% of the data was held as a test set. This seems to me too little compounds in the test set. Usually, it is kept with 20% of the whole data set.

Response: We thank the Reviewer for this valid point and agree that a 20% test set is the standard used in most chemoinformatic studies. We have devoted a significant computational effort to re-train the fold validations using a 80-20% data split. The text has been modified accordingly and the figures show the updated results.

* Moreover, did the authors use an external test set?

Response: The ZairaChem pipeline has been validated using external datasets from the Therapeutics Data Commons (<https://tdcommons.org>) benchmark. Our AutoML pipeline shows an excellent out-of-the-box performance, ranking between the 1st and 4th for various ADMET prediction tasks (Extended Data Table 2). For a more comprehensive explanation, please see our previous answer to this Reviewer.

* Datasets were very unbalanced (eg. ~15% actives to 85% of inactive). How did the authors deal with this? Did the authors try to balance the datasets?

Response: We thank the Reviewer for raising this very important point. Data imbalance is a well-known phenomenon in early-stage drug discovery (hit-to-lead), as one scopes out the chemical space around the minimum pharmacophore. This will of course be different in more 'mature' projects, in which robust structure-activity relationships have been established, and so, the relative proportion of active compounds is expected to increase as a project progresses. In any case, to clarify the imbalance in the datasets, we have added the percentage of active and inactive molecules in Extended Data Table 1. Furthermore, to address the imbalance issue we have added a combination of three balancing techniques,

including oversampling the positive class (K-Means SMOTE), undersampling the negative class (edited nearest neighbours) and a combination of over and under sampling with SMOTE-Tomek. We have used the imbalance-learn python library to that end. Methods have been updated accordingly.

* How were the activity cut-offs selected? Are there any rational why for Pf it was 0.1 uM and for Mtb it was 5 uM?

Response: Activity cut-offs were determined according to the H3D Centre screening cascade checkpoints and in consultation with relevant experts. Potency and efficacy values typically vary across disease areas and pathogens, in agreement with different target product profiles (TPPs). For example, therapeutic agents against *P. falciparum* are typically efficacious at nanomolar IC₅₀ *in vitro* potencies while, for *Mtb* – until more efficacious therapies are developed – much higher doses (in the micromolar range) are typically required for *in vitro* potency. Therefore, laboratory assays for these two disease areas use different cut-off values to determine if a molecule should progress in the cascade. Hence, the activity cut-off for *P. falciparum* is currently lower than for *M. tuberculosis*. To allow for flexible selection of compounds based on the assay criteria (typically, more restrictive in the advanced drug discovery stages) we provide not only the binarised (0: inactive, 1: active) outcome for the prediction but also the probability of belonging to the active class. The probability is a continuous value that allows independent researchers to set a threshold that suits their requirements (by default, the probability threshold is set at 0.5). We have made this clearer in the text.

* How were the descriptors chosen?

Response: ZairaChem contains a set of by-default descriptors, which is configurable via a simple configuration file where descriptors from the Ersilia Model Hub can be specified. Selection of descriptors were done following two criteria:

On one hand, given the broad range of tasks to be covered (from solubility prediction to whole-cell assay bioactivity, or interaction with drug-metabolising enzymes), we tried to cover the taxonomy of small-molecule descriptors available to the community. Thus, we selected Mordred (a comprehensive descriptor physicochemical properties), ECFP counts (representing the 2D topology), a graph-based transformer pre-trained on a large chemical space (Grover), a chemical language model (ChemGPT), and a bioactivity profile descriptor (Chemical Checker). We have now added another type of descriptor, the Ersilia Compound Embedding, as a representative of the transfer learning and few-shot learning strategies.

On the other hand, specific selection criteria within the relatively large marketplace of descriptors within these categories were based on (a) ease of implementation, (b) dimensionality, (c) computing speed, (d) popularity, and (e) observed overall performance across tasks, including previous expertise from our team.

Generally, we found the selected set of descriptors to be a reasonable choice, with none of them being consistently better, or worse, than the rest across tasks. Importantly, ZairaChem contains a “blending” meta-prediction step at the end where prediction results from each descriptor are aggregated in a weighted manner, depending on the task. This effectively upweights descriptors that are more performant for the task of interest. Also, please note that

within ZairaChem, dimensionality reduction is often performed to retain essential components of a descriptor array.

An improved explanation of descriptor choice has been added to the Methods section and is highlighted in yellow in the revised manuscript.

* Compounds seem to have many chiral centers. How did the authors deal with the stereochemistry of the compounds? Did the activity or biological endpoint was measured on the pure stereoisomers or racemic mixture?

Response: We thank the Reviewer for raising this point. Biological experiments and endpoints correspond to the annotations recorded in the H3D library, according to whether they are pure stereoisomers, a mixture of diastereomers, etc. This granularity in stereochemistry is subsequently lost during the MELLODY-Tuner standardisation protocol where different enantiomers collapse to the same flattened SMILES string. While we acknowledge that different stereochemistry may elicit different biological responses and that this is a limitation in the pipeline, the vast majority (>90%) of compounds in this work are achiral. Of the remaining chiral compounds, at least 50% are racemic mixtures. It is also noteworthy that in some cases, there is no activity difference between the racemic mixture and respective enantiomers, which explains why some antimalarial drugs such as chloroquine were developed as racemates for low cost of goods.

In addition, given the additional resource cost associated with synthesising pure enantiomers, drug discovery programs in the early stages of the pipeline prioritise making broad chemical modifications to establish the SAR/SPR trends in the chemical space of interest. Furthermore, chirality is of less concern for phenotypically-driven projects which form the majority of the H3D Centre's portfolio. It is within this context that the models have been developed and that we envisage the models in this work making the greatest impact toward accelerating drug development.

* Why did the authors chose to use both PCA and UMAP for analyzing the chemical space? this should be better explained and discussed.

Response: We believe these methods are complementary approaches to visualising chemical space. A PCA is a linear rescaling of the dataset that preserves the overall global relationship between data points (i.e. long-range or global distances) while a UMAP is a nonlinear rescaling that preserves local clustering between structurally-related compounds and provides better insight into the homogeneity of a dataset in chemical space. The current caption of Figure 3 indicates these differences and we have added a similar, succinct explanation to the methods section.

* Methods section: In my opinion, it is very general and brief and impossible to reproduce the work.

Response: We thank the Reviewer for pointing this out and agree with the Reviewer. We have improved the Methods section to include more information. Importantly, we have also extended the available online documentation of the ZairaChem pipeline (README file and official Ersilia code documentation).

* Authors should make all files, datasets and scripts available in a git hub so we could test the pipeline and all the community could benefit from this work.

Response: As noted above, in this revision we have put special emphasis in making our work available to the community. While there are intellectual property restrictions regarding the open sharing of H3D's proprietary chemical library, we have found a workaround to release ZairaChem models in 'anonymised' form (i.e. excluding any traces of small molecule structure data in the internal files stored by the model), and they are now available for download. In addition, and following a related comment by Reviewer 1, we have built a web-based interface (h3dscreening.ersilia.io [link]) to quickly run predictions on light models based on H3D data. ZairaChem scripts, and auxiliary repositories containing source code used in the screening, are all publicly available as part of Ersilia's GitHub organisation profile.

* how were the compounds curated? And how about the duplicates were analyzed? In case that the same compound was tested on two or more different assays, how was this analyzed and deal?

Response: The H3D chemical library is held in a database built on the Dotmatics software platform. All compounds in this database with registered experimental results were collected into raw datasets for each assay of interest, which spans compounds synthesised since the founding of the H3D Centre in 2010. Indeed, most compounds had replicate assay measurements, ensuring data reliability, that needed to be accounted for in order to standardise data to have a single label for each SMILES.

For each compound, first the variability between replicates was determined by calculating the mean of the differences for every pair of assay measurements, followed by taking this as a ratio of the overall mean measurement to calculate a relative error. All compounds that had a relative error >1 were discarded from the dataset, i.e. those compounds with highly variable differences between experimental data points that would cause an unreliable label (approximately 10% of all compounds). For the remaining compounds, the mean of the experimental values was assigned as the corresponding final label. This provided a flexible approach across virtual screening cascade assays whose outcomes spanned several orders of magnitude.

Compounds were further curated by selecting experimental values corresponding to a single set of assay conditions to reduce noise in the data. This was particularly relevant in the case of, for example, *Mtb*, in which cells exhibit differing growth characteristics under different cell culture conditions. The assay conditions and activity cut-offs were both selected in consultation with experimental scientists at H3D to ensure relevance of the resulting models.

We have expanded the methods section to include further discussion of the data curation as highlighted **in yellow** in the revised manuscript.

* For the benchmarking, other datasets should be used.

Response: We agree and, indeed, during this revision we have expanded our external validations substantially. As explained in an earlier response, we have benchmarked the ZairaChem pipeline using the Therapeutics Data Commons which provides ready-to-use datasets for easy benchmarking of AI/ML pipelines. Our models have ranked amongst the top three in all classification problems, except for BBB_Martins and CYP3A4_Veith (4th)

(Extended Data Table 2). In addition, to further address the Reviewer's comment, we have included additional benchmarking for *P. falciparum* and *M. tuberculosis*, as well as a selection of ADME assays, evaluating the performance of third-party models on H3D data and comparing it to the ZairaChem performance. Overall, we have found ZairaChem to be a state-of-the-art, fully automated pipeline that can be confidently applied across a broad range of tasks.

As instructed, we have electronically resubmitted a revised manuscript incorporating the various revisions as outlined above. We remain available to further edit/revise the manuscript as you may require. In the meantime, we trust that these revisions meet with your approval.

REVIEWERS' COMMENTS

Reviewer #1 (Remarks to the Author):

Authors provided substantial revisions to the manuscript that address reviewer's comments. They provided some examples of prospective analysis, developed a public tools for using 'light' versions of their models. These revisions are sufficient for publication.

Reviewer #2 (Remarks to the Author):

In this revised version of the manuscript, authors have improved their manuscript accordingly to the reviewers comments, specially improving the methods section, making all data available and more importantly, benchmarking their models with other previously published and applying their pipeline for a “real-world” example.

Moreover, they have devoted a significantly computational effort to fully accomplish most of all requests made by the two reviewers. The tone manuscript is now broader, to making it accessible and interesting to a wider audience.

So, I strongly recommend this paper for publication in Nature Communications.

Response to Reviewers' Comments

NCOMMS-23-01494-T: First fully-automated AI/ML virtual screening cascade implemented at a drug discovery centre in Africa

We'd like to thank the Reviewers for their positive feedback. We are grateful for the opportunity to publish our work at *Nature Communications*.

Reviewer #1 (Remarks to the Author):

Authors provided substantial revisions to the manuscript that address reviewer's comments. They provided some examples of prospective analysis, developed a public tools for using 'light' versions of their models. These revisions are sufficient for publication.

We'd like to thank the Reviewer for acknowledging the effort in addressing his/her comments and, especially, developing an easy-to-use interface that will increase the impact of our work.

Reviewer #2 (Remarks to the Author):

In this revised version of the manuscript, authors have improved their manuscript accordingly to the reviewers comments, specially improving the methods section, making all data available and more importantly, benchmarking their models with other previously published and applying their pipeline for a "real-world" example.

Moreover, they have devoted a significantly computational effort to fully accomplish most of all requests made by the two reviewers. The tone manuscript is now broader, to making it accessible and interesting to a wider audience.

So, I strongly recommend this paper for publication in Nature Communications.

We'd like to thank the Reviewer for his/her recommendation to publish our work in *Nature Communications*. We are indeed convinced that the improvements made following Reviewers' comments have increased the reach and interest of our manuscript.